# Exposure to images showing (non)adherence to physical distancing rules: Effect on adherence behavior and perceived social norms

Sanne Raghoebar[1]*, Joyce Delnoij[2], Bart A. Kamphorst[3], Henk Broekhuizen[4]

1 Department of Social Sciences, Consumption and Healthy Lifestyles Group & Education and Learning Sciences Group, Wageningen University and Research, Wageningen, The Netherlands, 2 Department of Social Sciences, Environmental Economics and Natural Resources Group & Urban Economics Group, Wageningen University and Research, Wageningen, The Netherlands, 3 Department of Social Sciences, Law Group & Philosophy Group, Wageningen University and Research, Wageningen, The Netherlands, 4 Department of Social Sciences, Health and Society Group, Wageningen University and Research, Wageningen, The Netherlands

* sanne.raghoebar@wur.nl

**Data Availability Statement:** All data files will be made available at the Open Science Framework (https://www.osf.io/uek2p).

## Abstract

### Introduction

Adherence to behavioral measures such as physical distancing are key to mitigating the effects of viral pandemics such as the COVID-19 pandemic. Adherence depends in part on people's perception of what others do (descriptive norms) or approve of (injunctive norms). This study examines the effects that exposure to images depicting people following or breaking physical distancing rules have on perceptions of descriptive and injunctive norms and subsequent adherence behavior.

### Methods

An online between-subjects experiment ($n = 315$) was conducted, in which participants were exposed to a set of five photographs of different public spaces in which people either did or did not adhere to physical distancing rules (pre-registration: https://www.osf.io/uek2p). Participants' adherence behavior was measured using a triangulation of measures (incentivized online behavioral task, vignettes, intention measure). Perceptions of relevant social norms were also measured.

### Results

Mann-Whitney tests showed no effects of condition on perceptions of descriptive and injunctive norms or on adherence behavior. Linear regressions showed that both component paths of the indirect effect (condition on norm perceptions, and norm perceptions on adherence behavior) were non-significant, hence mediation analyses were not conducted.

**Funding:** This work was conducted as part of the Health research programme of the Department of Social Sciences at Wageningen University and Research (WUR). The study itself was funded through a Research Costs Grant for postdoctoral researchers by the Wageningen School of Social Sciences (WASS grant number 21-046). The funders had no role in study design, data collection and analysis, decision to publish, or preparation of the manuscript.

**Competing interests:** The authors have declared that no competing interests exist.

**Abbreviations:** ANP, Algemeen Nederlands Persbureau, the largest news agency in the Netherlands; COVID-19, Corona Virus Disease 2019 caused by Severe Acute Respiratory Syndrome Coronavirus-2; OLS, ordinary least squares.

## Conclusions

Exposure to images of people following (compared to breaking) physical distancing rules did not affect adherence to such rules or perceived norms. We surmise that a single exposure to such images, especially in the context of COVID-19, is insufficient to affect behavior. We therefore recommend performing a comparable experiment in which participants are exposed repeatedly to images showing people (non)adhering to a specific behavior in a particular context for a longer period.

## Introduction

Behavioral measures such as physical distancing are believed to be key to mitigating the effects of viral pandemics such as the COVID-19 pandemic [1,2]. The extent to which the pandemic can be contained largely depends on the effectiveness of policies that aim to promote preventive health behavior among the public [3,4]. Since public support for such behavioral health policies may in turn depend in part on people's perception of what others do or (dis)approve of [5–7], it is crucial to understand how these perceptions are shaped.

One important factor in this regard is the exposure to media. The influence of media exposure on people's perceptions and behaviors has been widely investigated in different health domains (e.g., see [8–10]). Based on cultivation theory, repeated exposure to similar media content may result in people thinking the world's social realities are just as they are depicted in the media. This may subsequently influence their perceptions and behaviors [11,12]. The consistency of the message, together with exposure frequency, are important factors affecting the exact influence of such media exposure–the reason why the storytelling function of television is so powerful [13]. As a consequence, frequent media exposure may generate "misperceptions" of social reality, i.e. discrepancies between the world as depicted in media and the "real" world [14]. As such, media portrayals can have unintended consequences, in the sense that if people are (repeatedly) exposed to depictions of undesired behaviors, they may in effect be influenced to engage in undesired behaviors themselves [15]. In this context, consider how, during the COVID-19 pandemic, media outlets in the Netherlands frequently showed images of crowded places where people did not adhere to the communicated 1.5 meters physical distancing rule (e.g., see [16] or [17]). In line with cultivation theory, one may argue that repeated exposure to such images may give rise to the perception that most people are not adhering to physical distancing policies.

One mechanism through which frequent media exposure can shape a person's public image is by affecting perceptions of *social norms* [14,18,19]. According to cultivation theory, people can internalize social norms from media exposure by developing perceptions about the occurrence of the portrayed behavior. This also means that people may "misperceive" the objective prevalence of a behavior if the media exposure is not representative of the actual prevalence of the depicted behavior [14,19]. Following social norm theory [6], those social norm perceptions may be constructed through two conceptually and motivationally distinct processes [6,20]. First, people may form a distorted perception of the prevailing *descriptive norm*–a perception about what others in fact do–by "misperceiving" the actual prevalence of a behavior in a specific environment as a result of the media content to which they are exposed [6,14]. This perceived descriptive norm may subsequently affect their own behavior since people tend to follow the behavior of the majority (this is shown for a variety of different behaviors; for example, see [6,15,21]). Second, people may "misperceive" the prevailing *injunctive norm*–a perception about what should be done according to others–and form (mistaken)

beliefs about others' (dis)approval of certain behavior [6]. Presumably, exposing people to images of others' non-adherence (such as breaking physical distancing rules) may give people the idea that they have license to break the rules themselves, as this behavior to them seems to be socially acceptable, or at least have limited social sanctions (such as exclusion) [20]. *Vice versa*, an observation that other people mainly adhere to the communicated rules may lead to the perception that non-adherence to these rules is deviant in nature and will be (socially) sanctioned [14].

Preventive health behavior policies related to the COVID-19 pandemic (such as physical distancing) are highly collective in nature and their effectiveness is presumably strongly affected by perceptions of what others do and (dis)approve of. As media exposure may affect social norm perceptions, this paper is particularly focused on the influence of visual images used in media outlets. Specifically, we formulate the following hypotheses:

**Hypothesis 1.** Visual exposure to images depicting other people adhering (versus non-adhering) to physical distancing rules will increase adherence (versus non-adherence) to such rules.

**Hypothesis 2.** Visual exposure to images depicting other people adhering (versus non-adhering) to physical distancing rules will increase perceptions of

a. descriptive norms suggesting that other people generally adhere (versus non-adhere) to such rules, and

b. injunctive norms suggesting that physical distancing rules ought to be adhered (versus non-adhered) to according to other people.

**Hypothesis 3.** The effect of visual exposure to images depicting other people adhering (versus non-adhering) to physical distancing rules on adherence (versus non-adherence) to such rules is mediated by perceptions of descriptive norms and injunctive norms.

The hypotheses were tested in an online between-subject experiment conducted in the Netherlands, where participants were exposed to a set of photographs depicting people either following (desired behavior) or breaking (undesired behavior) physical distancing rules.

## Methods

### Design

A between-subject design with two conditions was performed in an online study. Such a design minimizes the risks of learning, carry-over, experimenter demand, and boredom effects. Participants were exposed to a set of five similarly-sized photographs of different public spaces in which more than three people were present. The adherence to 1.5 meters physical distancing rules was manipulated between two conditions: one group was shown photographs depicting people not adhering to 1.5 meters physical distancing rules (non-adherence condition), the other group was shown photographs depicting people adhering to such rules (adherence condition). Subsequently, participants' adherence behavior was measured using a triangulation of measures: an online behavioral task in the form of an incentivized game, exposure to vignettes describing hypothetical dilemmas regarding adherence to physical distancing rules and a single item intention measure. Finally, perceptions of descriptive social norms and injunctive social norms regarding adherence behavior were measured.

### Participants and sample size

Dutch adults (aged 18 up to and including 65 years) were recruited through the online survey platform Prolific Academic. To determine the sample size, a Monte Carlo power analysis for

indirect effects was performed [22]. This shows that a power of 0.80 ($p < 0.05$) is reached with 294 participants in a model with two parallel mediators (perceived descriptive norms and perceived injunctive norms). Based on the correlations found in previous research [23,24], a correlation of $r = 0.30$ was assumed between the independent variable and the mediators, and the outcome variable. Further, a correlation of $r = 0.30$ was assumed between the mediators and the outcome variable, and a correlation of $r = 0.40$ was assumed between the mediators themselves. It was planned to exclude participants who correctly reported the study aim, therefore the sample size was set to 320 participants (approximately 160 participants per condition).

To ensure that incentives in our experiment were aligned with decision making in everyday life, participants' compensation in part depended on their choices in the incentivized game [25]. Hence, participants received a fixed payment of €1.74 as well as a bonus payment based on their performance, which varied between €0 and €3.03 ($M = $€2.10$, SD = 0.43$)

## Materials (photographs)

Photographs depicting people in public spaces were used for the exposure manipulation. Selection of the photographs was determined by the results of an online pilot study in the form of a rating task. In total, 59 Wageningen University and Research employees gave their written consent to participate in the pilot study (77.19% female; age $M = 38.61$, $SD = 12.97$). Participants were exposed to different photographs sourced from the ANP (Algemeen Nederlands Persbureau, large news agency in the Netherlands) and editorial photographs from Shutterstock. The photographs were made during the COVID-19 pandemic, and ranged in (a) the type of public space depicted on the photograph (shopping streets, parks, supermarkets (outside and inside) and beaches), (b) the number of people present and (c) the adherence to 1.5 meters physical distancing rules. Pilot participants were instructed to rate the different photographs on adherence to 1.5 meters physical distancing rules ("To what extent do people in this photo adhere to the physical distancing rules of 1.5 meters?"). This item was rated on a scale from 1 (not at all) to 7 (very much). The presentation order of the photographs was evenly randomized across participants. Five photographs that scored high on adherence to 1.5 meters physical distancing were selected for inclusion in the adherence condition and five photographs that scored low on adherence to 1.5 meters physical distancing were selected for inclusion in the non-adherence condition (Table 1).

## Procedure

After the photographs had been selected, the main study was pilot tested on the online survey platform Prolific Academic ($n = 40$). As small adjustments had to be made afterwards (e.g., to the explanation of the behavioral task), data from these participants were not included in the final sample.

For the main survey, which was administered using Qualtrics, participants were invited to participate in a study about decision making in times of uncertainty. They were included if

**Table 1. Selected images with mean and standard deviation for the adherence score as asked in the image pilot study.**

| | Adherence condition | | | Non-adherence condition | | |
|---|---|---|---|---|---|---|
| | Context | Mean | SD | Context | Mean | SD |
| Photo 1 | Park | 5.82 | 1.62 | Park | 1.09 | 0.34 |
| Photo 2 | Shopping street | 6.07 | 1.19 | Shopping street | 1.37 | 0.56 |
| Photo 3 | Beach | 5.93 | 1.13 | Beach | 1.67 | 0.87 |
| Photo 4 | Supermarket: outside | 5.44 | 1.41 | Supermarket: outside | 1.95 | 1.16 |
| Photo 5 | Supermarket: inside | 6.28 | 0.94 | Supermarket: inside | 2.70 | 1.30 |

they used a desktop or laptop computer. Once participants had given their written informed consent, they were exposed to a set of five photographs, showing either adherence or non-adherence to 1.5 meters physical distancing rules depending on the assigned condition. The photographs were presented in the order as shown in Table 1, on separate pages. For each photograph, participants were required to describe the photograph in as much detail as possible, using at least 30 words. Also, they were asked to evaluate five statements about the photographs that were unrelated to physical distancing measures (e.g., "The shown photograph is colorful" and "The shown photograph was taken on a hot day"), rated on a scale from 1 (totally disagree) to 7 (totally agree).

After the exposure phase, participants were presented with instructions for the behavioral task. In this task, which was adapted from Kimbrough & Vostroknutov [26], participants were instructed to move a stick figure through a supermarket. In doing so, each participant faced five decision moments where they could either choose to move past another shopper in close proximity or wait for them to move out of the way (for details, see the Measures section). Thereafter, participants completed the proposed mediator items. Each item was presented on a separate page in an evenly randomized order. Participants were then exposed to two different scenarios, each presented on a separate page in an evenly randomized order. For each vignette, participants were instructed to complete the two items related to the vignettes. Participants then completed the intention item, the two outcome expectancies items and the risk perception item.

Subsequently, participants were presented the items related to COVID-19 infection, COVID-19 vaccination, and their use of digital contact tracing apps. Then, participants reported any changes in occupation due to COVID-19 as well as their demographic information (sex, age, municipality of residence, education and nationality). Afterwards, they completed the awareness of the study aim question as an exclusion criterion, and completed a manipulation check to assess their recall of the images to which they were exposed. Finally, participants were thanked, debriefed and reimbursed.

## Measures

**Outcome variables.** To approximate adherence behavior to the 1.5 meters physical distancing rule, a triangulation of measures was used: an online behavioral task (an incentivized game measuring waiting time and number of shoppers waited for) and two different intention to adhere measures, including vignettes and a single item intention measure [27] that can be found in Table 2.

**Online behavioral task.** In the online behavioral task all participants were asked to move through a busy supermarket in which they faced five choice situations where they could either

**Table 2. Items assessing intention, perceptions of descriptive norms, and injunctive norms.**

| Variable | Items |
| --- | --- |
| Intention to adhere | "To what extent do you intend to adhere to the 1.5 meters physical distancing rule?" (1 (not at all) to 7 (very much)) |
| Perceptions of descriptive norms | 1. "How likely is it that other participants of this research would adhere to the 1.5 meters physical distancing rule?" (1 (not at all likely) to 7 (extremely likely))<br>2. "How likely is it that most participants of this research would adhere to the 1.5 meters physical distancing rule?" (1 (not at all likely) to 7 (extremely likely)) |
| Perceptions of injunctive norms | 1. "To what extent do other participants of this research think you ought to adhere to the 1.5 meter physical distancing rule?" (1 (not at all) to 7 (very much))<br>2. "To what extent do other participants of this research think that it is appropriate that you adhere to the 1.5 meter physical distancing rule?" (1 (not at all) to 7 (very much)) |

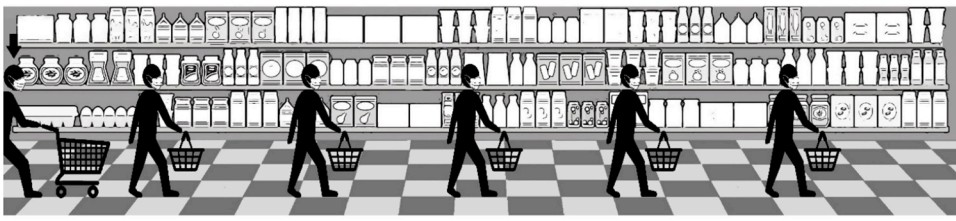

**Fig 1. Screenshot of the behavioral task in the supermarket context.** The word "lopen" on the button is Dutch for "walk".

follow or break the physical distancing rule of 1.5 meters. As we will explain below, adherence to this rule was costly for participants in terms of monetary reward. In the original rule-following task [26]–which is well established in the literature (e.g., see [26,28])–participants must decide whether or not to wait for a traffic light to turn green. In our behavioral task participants moved through a supermarket and had to decide whether to wait for other shoppers to move out of the way, thus keeping the required minimal physical distancing of 1.5 meters, or pass these shoppers without complying with the physical distancing rule of 1.5 meters. Fig 1 shows a screenshot of the task.

Participants controlled the stick figure with the shopping cart who walked from the left to the right side of the screen. Participants could start the task by clicking anywhere on the screen, after which the figure would start walking towards the first shopper. The figure automatically stopped at 1.5 meters of physical distance from the first shopper it encountered. This shopper moved out of the way after five seconds. However, participants were free to press a button labeled WALK ("LOPEN") any time after the stick figure stopped, which would force the stick figure to bypass the other shopper within 1.5 meters (and thus breaking the physical distancing rule of 1.5 meters). When a participant pressed WALK, the stick figure continued to walk to the right until it encountered the next shopper, at which point the stick figure automatically stopped at 1.5 meters of physical distance again. At this point, the participants could again choose between waiting or passing. Consistent with the original task, there were five shoppers in total (thus five choice situations).

Participants started with an endowment of €4, and were informed that for each second they spent in the task, €0.04 was subtracted from this endowment. In the instructions for this task, participants were informed that "The rule is to wait until it is possible to keep 1.5 meters physical distancing from other customers." Three minor changes to the task instructions were made after reviewing the results of a pilot study ($n = 40$). First, it was made explicit that participants could choose to wait or pass the other customers. Second, after communicating the rule, it was stated that it was up to the participant to decide how to deal with the rule. Third, orange circles were included in the screenshots of the task in the instructions, to clearly indicate which stick figure was controlled by the participant. No other information, apart from the payment scheme and a general description of the walking procedure along with screenshots, was provided in the instructions. The full instructions for the task can be found in the Supporting information under Methods.

Adherence behavior was captured by measuring the time participants waited at each choice situation (seconds) and the number of shoppers that participants waited for (ranging from 0–5).

**Intention to adhere: Vignettes.** Apart from the behavioral task, participants were presented with a vignette-based outcome measure designed to capture how they intend to behave in a certain situation. Two different vignettes were presented to participants, the first describing a person (Robin) who adheres to the 1.5 meters physical distancing rule, and the second describing a person (Chris) who does not adhere to the 1.5 meters physical distancing rule. Robin and Chris are gender neutral names that are common in the Netherlands. Participants were asked to read each vignette carefully and then evaluate two items on a scale from 1 (totally disagree) to 7 (totally agree): "If I were Robin (Chris) I would have done the same" and "When I find myself in a similar situation, I will not act in the same way as Robin (Chris)". After reverse-coding the appropriate items, a mean score for the four items was calculated to measure intention to adhere to the 1.5 meters physical distancing rule (Cronbach's $\alpha$ = 0.77).

Both vignettes were tested in a pilot study among 59 Wageningen University and Research employees (as part of the same pilot study as presented in the Materials section). Based on their feedback, minor grammatical adjustments were made to the texts. The translated vignettes (from Dutch) are presented below.

Vignette 1: *"It is 5:30 pm and Robin is walking to the supermarket to get dinner. Once arrived at the supermarket, it turns out to be very busy at the entrance: there is a line of customers outside the store waiting to grab a shopping cart or shopping basket. Robin sees a number of people quickly slip past the line to enter the store. In doing so, people are not keeping 1.5 meters distance. Given the crowd, it seems difficult to Robin to keep enough distance in the store. Robin decides to return home to make a meal out of leftovers."*

Vignette 2: *"It is 11:45 am on a Monday and Chris is in the supermarket buying groceries for the next few days. Upon arrival it was relatively quiet in the store, but now it is getting busier. Nearing the checkout, Chris realizes that the bread is missing. Chris walks back to the bread department and finds a crowd of people walking closely past each other to grab lunch. Chris puts the shopping cart aside, moves quickly between the people and shopping carts without keeping a distance of 1.5 meters, snatches a loaf of bread from the shelf, and then makes their way to the checkout as quickly as possible."*

**Mediator variables.** Perceptions of descriptive and injunctive norms were both measured with two items as listed in Table 2 (items were inspired by [23,24]). A mean score for the two items of descriptive norms was calculated using the Spearman-Brown formula (predicted reliability = 0.87). The items of perceived injunctive norms were separately analyzed using the same formula, as reliability showed a poor level of internal consistency (predicted reliability = 0.68).

**Descriptive variables.** Outcome expectancies were measured by two items taken from [27]: "If I adhere to the measures on coronavirus, I have a lower risk of getting infected" and "If I adhere to the measures on coronavirus, fewer people will get infected". Both of these were rated on a scale from 1 (totally disagree) to 7 (totally agree). A mean score for the two items was calculated using the Spearman-Brown formula (predicted reliability = 0.87).

Risk perception was measured by one item taken from [27]: "I have little chance of getting infected with the coronavirus". It was rated on a scale from 1 (totally disagree) to 7 (totally agree).

Changes in occupation due to COVID-19 were measured. Answer options included (multiple answers possible): "No", "Yes, I am or have temporarily been unable to work due to restrictions", "Yes, I am or have temporarily been unable to work due to infection with COVID-19", "Yes, work pressure has increased" and "Yes, I work from home more often". A dichotomous variable (yes/no) was created.

Participants were asked whether they had ever had COVID-19. Answer options included: "Yes, in the last half year", "Yes, longer than half a year ago", "No" and "I do not want to answer this question". A dichotomous variable (yes/no) was created.

Participants were asked whether they know someone who is (or has been) infected with COVID-19. Answer options included (multiple answers possible): "Yes, immediate family", "Yes, extended family", "Yes, friends", "Yes, acquaintances", "No", and "I do not want to answer this question". A dichotomous variable (yes/no) was created.

Participants were asked about their COVID-19 vaccination status. Answer options included: "Yes, I am completely vaccinated", "Yes, I am partly vaccinated" (i.e. for vaccines that requires multiple shots for full effectiveness), "No, not yet, I will get my vaccination later", "No and I do not intend to get vaccinated", and "I do not want to answer this question". A dichotomous variable (yes/no) was created.

Usage of a (voluntary) Dutch digital contact tracing app was measured. Answer options included: "Yes, it is always turned on", "Yes, when I think about it", "No" and "I do not want to answer this question". A dichotomous variable (yes/no) was created. As a follow-up question, participants who indicated to use the app were asked whether they have ever received a notification of the Dutch digital contract tracing app (yes/no).

Dichotomous variables were made for sex (male), nationality (Dutch), and educational level (academic education). The participant's municipality of residence was used to indirectly measure infection risk (measured by mean centering the average cases per 1000 inhabitants over the month June 2021).

**Manipulation check.**   Participants were shown ten photographs used in the exposure phase, presented in a random order. The included photographs were a combination of photos used in both the adherence and non-adherence condition. Participants were instructed to select those photographs that they had been exposed to. Participants who selected at least three correct photographs were coded as having passed the manipulation check.

*Exclusion criterion for analysis*. Participants' awareness of the study aim was assessed by asking the open-ended question: "What do you think the purpose of this research was?" Participants who explicitly linked the manipulation to the outcome variables and/or mediating variables were denoted as having correctly identified the study aim and were excluded from the analytic sample (as independently rated by two co-authors).

## Data analysis

The analyses were performed using Stata (version 16). Mann-Whitney tests (for continuous, non-normally distributed variables) and Pearson chi-square tests (for dichotomous variables) were performed to check significant differences between conditions in descriptive and outcome variables. We estimated ordinary least squares (OLS) regressions with and without relevant covariates (including descriptive variables and norm perceptions) for outcome variables to check significant differences between conditions. For mediating variables, we first estimated OLS regressions to check significant differences between conditions. Then, to test the mediating effect of social norms, first the conditions for mediation were checked by examining whether both components of the indirect effect were significant. OLS regressions were conducted to examine the effect of condition on each potential mediating variable and the effect of each potential mediating variable on each outcome variable. Had the conditions for mediation been met (thus showing significant effects for both component paths), we would have used Hayes's PROCESS macro to test for mediation (model 4) [16]. For more details, see our pre-registration at the Open Science Framework (https://www.osf.io/uek2p). Finally, explorative moderation analyses were performed.

## Results

### Participant characteristics

In total, 346 participants on Prolific Academic participated in the experiment, of which 315 participants were included in the analytic sample after planned exclusions (see S1 Fig in S1 File)–the included participants had a mean age of 28.27 years and 183 were male. The descriptive statistics per condition are reported in Table 3. For more detailed descriptions of the total participant sample, see S1 Table in S1 File.

The randomization check was conducted to check whether participant characteristics were comparable across conditions, showing a significant difference in nationality between conditions (see Table 3). Hence, primary analyses were repeated by including nationality in the model, but the results were not significantly impacted by the inclusion of this covariate. In the following sections we therefore reported the results without inclusion of this covariate.

### Manipulation check

All participants (315/315, 100%) correctly identified the photographs they were shown in the exposure phase–meaning that they correctly selected at least three out of five photographs they were exposed to–and thus all participants passed the manipulation check. Two participants in the adherence condition and one participant in the non-adherence condition incorrectly identified one photograph (and thus correctly identified four photographs). No significant differences were observed between the conditions in correct identification of the photographs, as shown by a Fisher's Exact test, $p = 1.00$.

**Table 3. Descriptive statistics per condition ($n = 315$), test statistic and $p$-value refer to non-parametric Mann-Whitney tests and Pearson chi-square tests.**

| | Adherence condition ($n = 160$) [b, c] | Non-adherence condition ($n = 155$) [d, e, f] | Test statistic | $p$-value |
|---|---|---|---|---|
| | Mean (SD) or Number (%) | Mean (SD) or Number (%) | | |
| **Differences between conditions in demographic information** | | | | |
| Age (y) | 28.38 (9.50) | 28.16 (9.59) | $U = 0.30$ | 0.77 |
| Sex (male) | 85 (53.1%) | 98 (63.2%) | $X^2(1) = 3.30$ | 0.07 |
| Nationality (Dutch) | 142 (88.8%) | 124 (80.0%) | $X^2(1) = 4.59$ | 0.03 |
| Education (academic education) | 65 (40.6%) | 59 (38.1%) | $X^2(1) = 0.22$ | 0.64 |
| Incidence rate in municipality | -0.02 (0.99) | 0.02 (1.01) | $U = -0.23$ | 0.82 |
| **Differences between conditions in descriptive variables related to COVID-19** | | | | |
| Outcome expectancies [a] | 5.72 (1.42) | 5.84 (1.26) | $U = -0.39$ | 0.70 |
| Risk perception [a] | 4.41 (1.77) | 4.53 (1.76) | $U = -0.59$ | 0.56 |
| Changes in occupation (no) | 55 (34.4%) | 57 (36.8%) | $X^2(1) = 0.20$ | 0.66 |
| Infection with SARS-CoV-2 | | | | |
| Personal (yes) | 21 (13.2%) | 22 (14.5%) | $X^2(1) = 0.11$ | 0.75 |
| Surroundings (yes) | 142 (89.3%) | 143 (92.9%) | $X^2(1) = 1.21$ | 0.27 |
| COVID-19 vaccination status (yes) | 81 (51.3%) | 81 (53.6%) | $X^2(1) = 0.18$ | 0.68 |
| Usage of Dutch digital contact tracing app (yes) | 46 (29.1%) | 49 (31.6%) | $X^2(1) = 0.23$ | 0.63 |

[a] Ranging from 1–7.

[b] $n = 159$ for infection with COVID-19 (personal and surroundings) because of refusal to answer.

[c] $n = 158$ for COVID-19 vaccination status and usage of a digital contact tracing app because of refusal to answer.

[d] $n = 154$ for age because of missing values and for infection with COVID-19 (surroundings) because of refusal to answer.

[e] $n = 152$ for infection with COVID-19 (personal) because of refusal to answer.

[f] $n = 151$ for COVID-19 vaccination status because of refusal to answer.

**Table 4. Adherence behavior and the proposed mediators per condition (n = 315), test statistic and p-value refer to non-parametric Mann-Whitney tests.**

|  | Adherence condition (n = 160) | Non-adherence condition (n = 155) | Test statistic | p-value |
|---|---|---|---|---|
|  | Mean (SD) or Number (%) | Mean (SD) or Number (%) |  |  |
| **Effect of condition on adherence behavior** |  |  |  |  |
| Online behavioral task |  |  |  |  |
| Waiting time (seconds) | 23.32 (11.35) | 24.38 (13.65) | U = -0.90 | 0.37 |
| Number of shoppers waited for [a] | 3.62 (1.96) | 3.79 (1.85) | U = -0.94 | 0.35 |
| Intention to adhere |  |  |  |  |
| Vignettes [b] | 4.12 (1.56) | 4.21 (1.62) | U = -0.55 | 0.59 |
| Single item measure [b] | 5.71 (1.27) | 5.79 (1.24) | U = -0.71 | 0.48 |
| **Effect of condition on perceptions of descriptive norms and injunctive norms** |  |  |  |  |
| Perceptions of descriptive norms [b] | 4.04 (1.58) | 3.93 (1.70) | U = 0.60 | 0.55 |
| Perceptions of injunctive norms |  |  |  |  |
| Item 1 [b] | 4.96 (1.52) | 4.74 (1.78) | U = 0.82 | 0.41 |
| Item 2 [b] | 5.28 (1.45) | 5.07 (1.58) | U = 1.11 | 0.27 |

[a] Ranging from 0–5.

[b] Ranging from 1–7.

## Adherence behavior (hypothesis 1)

No significant differences between conditions were found in the online behavioral task regarding the time participants waited at each choice situation and the number of shoppers that participants waited for as a proxy for their adherence to the 1.5 meters physical distancing rule. Similarly, there were no significant differences between conditions in intention to adhere–as measured by the vignettes and the single item measure (Table 4). For robustness checks, see S2 Table in S1 File. The effect of treatment remained non-significant with inclusion of covariates.

## Perceptions of descriptive norms and injunctive norms (hypothesis 2 and 3)

No significant differences between conditions were found in perceptions of descriptive norms and perceptions of injunctive norms (for both items), though the scores were in the hypothesized direction (Table 4). The results imply that the conditions for mediation were not met (see S3 Table in S1 File), hence mediation analyses were not performed.

## Moderation

To better understand the mechanisms behind our non-significant results, we explored the role of outcome expectancies and risk perception as potential moderators. Neither outcome expectancies nor risk perception appeared to moderate the relationship between the treatment and any of the outcome or mediating variables.

## Discussion

This paper examined whether visual exposure to images depicting other people adhering (versus non-adhering) to physical distancing rules affected adherence behavior. We hypothesized that this relationship could be explained through people's perceptions of what others do (descriptive norms) and perceptions of what people think others approve of (injunctive norms). A triangulation of measures (an incentivized game, vignettes, and a single item intention measure) was used to capture Dutch people's adherence to the communicated 1.5 meters

physical distancing rules that were in force at the time of data collection [29]. Contrary to our expectations (hypothesis 1), the results demonstrate that visual exposure to images depicting other people following (versus breaking) physical distancing rules did not directly impact participants' adherence behavior. Further, the exposure manipulation did not affect participants' perceptions of descriptive norms about the typical adherence behavior of others towards physical distancing, nor did it affect perceptions of injunctive norms about the appropriateness of adherence to such rules according to others (hypotheses 2a and 2b, respectively). Consequently, no significant mediating role for perceptions of descriptive and injunctive norms was found in the relationship between the exposure manipulation and adherence behavior (hypothesis 3).

These findings do not align with previous studies showing that visual exposure to a stimulus (e.g., food portion size images or body size images) can affect people's norm perceptions [30,31]. For example, work by Robinson and colleagues [31] demonstrated that visual exposure to images showing either large or small food portions may affect people's perceptions of what constitutes a normal portion of that food. Similarly, then, we expected that exposure to our manipulation would have affected people's norm perceptions about physical distancing. Given the current results, it may be suggested that exposure to such photographs is not as influential in affecting participants' judgements about the normality and appropriateness of physical distancing behavior as we predicted. In what follows, we discuss several possible explanations for our non-significant results in more detail.

Reasoning from cultivation theory, a possible explanation may be that a *single* image exposure occasion at one point in time is insufficient to shape or influence social norm perceptions and behaviors (related to hypothesis 1 and 2). We designed and pilot tested the exposure manipulation to ensure that participants were exposed to relevant images for a relatively substantial duration (Table 1). Nevertheless, the present results may indicate that longer or more frequent exposure is required to change people's perceptions of social norms in this domain. Such reasoning is in line with work on television exposure that suggests that frequent (or greater) exposure to similar images on television may in the longer term lead to people constructing (more) congruent perceptions and behaviors [11,12,32]. Given this hypothesis, for future work we recommend performing a comparable experiment in which participants are exposed repeatedly to a set of images showing people (non)adhering to a specific behavior in a particular context for a longer period.

Reasoning from social norm theory [6], it is known that people typically conform to injunctive norms to build and maintain social relationships with relevant others, thereby increasing (limiting) the chance of social approval (sanctioning) [20]. Inferring injunctive norms thus requires participants to make an inference from the situations as depicted in the images to the situations in the ensuing experimental tasks. Given the non-significant results of the injunctive norm measure, it is a distinct possibility that participants did not engage in this kind of deliberation about potential social approval or disapproval after having been exposed to the images.

Another possible explanation from social norm theory pertains to the reference group as described in the descriptive and injunctive social norm measures of this study. The reference group was specified as 'other participants of this research', but it is questionable whether people would have sufficiently identified with the people in this reference group, and may therefore not have cared enough about these individuals' behaviors or their judgments (i.e. what they do or (dis)approve of; related to hypothesis 2a and 2b) to let it influence their own behavior [33]. Instead, the social environment in which participants performed the experiment (e.g., alone or in the presence of others) may have influenced their perceptions and behaviors. A question for further research would then be how the social environment interacts with the exposure manipulation (i.e. the photo task).

A different possible explanation of the present results pertains to the potential presence of strong pre-intervention perceptions of social norms. Participants had presumably been exposed to a wide variety of (inconsistent) behaviors, opinions, and expectations about physical distancing behavior prior to the experiment. For example, in the preceding months leading up to the present study, the topic of COVID-19 was extensively covered in the media [34]. This may have shaped pre-existing social norm perceptions and consequently may have minimized or nullified the effects of this study (related to hypothesis 1–3). To illustrate, consider how those who already strongly believed that adherence to physical distancing rules is common and appropriate may have been less easily influenced by photographs depicting other people non-adhering to such rules. This potential explanation of the results is supported by the observation that–irrespective of the manipulation–perceptions of descriptive norms and injunctive norms were correlated with physical distancing behavior (see S2 and S4 Tables in S1 File). In addition, recent work has demonstrated that norm perceptions regarding physical distancing consistently predicted physical distancing intentions [35]. Future research should therefore measure perceptions of social norms and the extent to which the participants are exposed to COVID-19 in media prior to the intervention in order to control for their potential influence.

The explanations posited above suggest that the photograph manipulation from the present study was not strong enough to significantly influence norm perceptions (hypotheses 2a and 2b) and subsequent behaviors (hypothesis 1 and 3). If this is indeed the case, this could be viewed as an interesting result in itself with respect to more general questions about the role of imagery in news coverage and health behavior communications, and the potentially limited influence images exert on the uptake of public health policies. This seems especially interesting from the perspective that news coverage has the potential to undermine new policies by highlighting undesirable behaviors (such as breaking physical distancing rules) and thereby indicating that this behavior is common [36]. Notably, the present results are in striking contrast to another recent, comparable study in the context of COVID-19 on the effects of written narratives on risk behavior and patience in economic games. In this study, Harrs, Müller & Rockenbach [37] found that the phrasing of a written narrative–optimistic, pessimistic, or balanced–affected how risk averse and patient participants were in the context of the economic games. Moreover, previous studies have shown that exposure to radio can also impact norm perceptions (e.g., see [38] or [39]). Thus, it may be that one-time exposure to images, at least without associated written or spoken context, is not as powerful at changing norm perceptions as other forms of media exposure. However, firm conclusions in this direction should be curtailed until more research has been conducted into the effects of images as a communication vehicle for norms related to public health (e.g., in comparison with textual and verbal communication).

Further research in this direction is also welcomed in light of several methodological limitations to the present study. First, it may be that the results were influenced by the epidemiological context around the time the study was conducted. During data collection, incidence rates in the Netherlands were relatively low and vaccination coverage was increasing rapidly [40]. It may thus be surmised that participants may have been less worried about contracting COVID-19 at the time of their participation than at the beginning of the pandemic, which may have affected how they performed in the behavioral task. Given the unpredictability of pandemics, it would be interesting to conduct a repeated survey over a longer time (assuming one or more pandemic waves occur over that time) and conduct a within-respondent analysis of the impact of epidemiological context on results.

Second, it could be that the task of actively describing elements from each photograph with a certain number of words may have made the physical distancing behavior of the people

depicted in the photographs less salient (i.e. focal in attention) to the participants. If this were indeed the case, it may explain the reported results because, from a social norm perspective [6], it is a prerequisite that social norms are salient in order to be able to exert an influence on behavior. However, we do note that the procedure of having participants perform an unrelated task in order to ensure a substantial exposure duration to the manipulation (although the exact duration was not pre-defined) was successful in other research [31]. Moreover, and further speaking against this type of explanation, we found that all participants correctly recalled the photographs they were exposed to (and thus passed the manipulation check).

Thirdly, one may reason that the included hypothetical behavioral measures may not have been adequate proxies for the behavior we were actually interested in, as none of them involved a real threat of infection. Given the ethically problematic nature of manipulating actual (real life) adherence behavior to physical distancing during the COVID-19 pandemic, the present approach offers a relatively robust way of approximating physical distancing behavior in a relatively safe manner. As such, we therefore consider the triangulation of measures used to estimate physical distancing behavior (including the online behavioral task) a strength of the present study. Still, since none of the measures involved a real threat of infection, it may be that participants' responses to these measures deviate from what their actual behavior would have been. That said, prior meta-analyses suggest that behavioral intention items do typically correlate with actual behavior [35].

Finally, there is the possibility that the non-significant results for our hypotheses are related to issues with the sample. For example, it could be that the power of the sample was too low to detect a true effect with a worthwhile effect size (cf. Lakens [41]). Though this possibility cannot be ruled out, the fact that our power analysis was based on estimations from prior research related to social norms reduces the plausibility of this explanation. Likewise, it cannot be ruled out that key characteristics of the entire sample, such as people's attitudes towards adherence of COVID-19-related norms, are not representative of the population, and that an effect would have been found if the sample had been constituted differently. It could be hypothesized, for example, that conducting the study entirely online–and thus allowing participants to partake from the safety of their homes¬–would have introduced an inclusion bias in favor of people who were already strongly oriented towards adherence of physical distancing norms. On the one hand such a suggestion seems to be strengthened by the relatively high mean score for outcome expectancies indicating that the included participants believed in the effectiveness of the measures on the coronavirus. On the other hand, nationwide Dutch adherence surveys indicate that a majority of the population supported (and tried to follow) the physical distancing rules that were in place at the time of the survey [42], suggesting that the high mean outcome expectancy was to be expected in a sample of the Dutch population. It would be interesting for future research to include a participant sample that varies in the extent to which they believe in the effectiveness of the measures on the coronavirus–e.g., by recruiting participants at different physical locations across the Netherlands. It may be hypothesized that outcome expectancies moderate the effect of the manipulation on norm perceptions and physical distancing behavior, in that the manipulation showing images of other people adhering to physical distancing rules may be more effective among a subgroup of participants believing in the effectiveness of such rules [14,27].

## Conclusion

Although the effect of visual exposure to a stimulus on norm perceptions has been shown repeatedly in different contexts and situations, this is, to our knowledge, the first study that examined the effect of exposure to imagery on adherence behavior in the context of COVID-19 –specifically by exposing people to images showing other people either adhering or non-

adhering to communicated physical distancing rules. The outcomes of the present study were not in line with our hypotheses, as exposure to images of people following (compared to breaking) physical distancing rules did not affect adherence to such rules (hypothesis 1) or perceived descriptive (hypothesis 2a) and injunctive norms (hypothesis 2b) regarding adherence behavior. However, if the non-significant results are indicative of a relatively weak influence of such imagery on adherence behavior and norm perceptions, then this may be an important insight for how health behaviors are best communicated. Further, we surmise that if the main reason for our non-significant results is the potential information overload that people may have experienced in relation to COVID-19 or the insufficiency of single exposure to such images to affect presumably rather stably established perceptions and behaviors, then we would recommend performing a comparable experiment in which participants are exposed repeatedly to a set of images showing people (non)adhering to a specific behavior in a particular context for a longer period.

## Supporting information

**S1 File. S1 Methods: Translated instructions online behavioral task.** S1 Results: S1 Fig. Participant flowchart. S1 Table. Detailed descriptions of the participant sample ($n = 315$). S2 Table. Robustness analyses ($n = 315$). We repeated the analyses for number of shoppers using ordered logit regressions. S3 Table. Component paths of the indirect effect of condition on adherence behavior for perceptions of descriptive norms and perceptions of injunctive norms ($n = 315$). S4 Table. Pairwise correlations ($n = 315$).
(DOCX)

## Acknowledgments

We would like to thank the survey participants for their contributions.

## Author Contributions

**Conceptualization:** Sanne Raghoebar, Joyce Delnoij, Bart A. Kamphorst, Henk Broekhuizen.

**Data curation:** Joyce Delnoij.

**Formal analysis:** Sanne Raghoebar, Joyce Delnoij.

**Funding acquisition:** Sanne Raghoebar, Joyce Delnoij, Bart A. Kamphorst, Henk Broekhuizen.

**Investigation:** Sanne Raghoebar, Joyce Delnoij, Bart A. Kamphorst, Henk Broekhuizen.

**Methodology:** Sanne Raghoebar, Joyce Delnoij, Bart A. Kamphorst, Henk Broekhuizen.

**Project administration:** Sanne Raghoebar, Joyce Delnoij.

**Software:** Joyce Delnoij.

**Visualization:** Sanne Raghoebar.

**Writing – original draft:** Sanne Raghoebar, Bart A. Kamphorst.

**Writing – review & editing:** Joyce Delnoij, Bart A. Kamphorst, Henk Broekhuizen.

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
