## [Decision Letter · Decision Letter 0]

25 Jul 2022

PONE-D-22-17082Exposure to images showing (non)adherence to physical distancing rules: Effect on adherence behavior and perceived social normsPLOS ONE

Dear Dr. Sanne Raghoebar,

Thank you for submitting your manuscript to PLOS ONE. After careful consideration, we feel that it has merit but does not fully meet PLOS ONE’s publication criteria as it currently stands. Therefore, we invite you to submit a revised version of the manuscript that addresses the points raised during the review process.

As you can see below, we received two reports commenting on your paper. Both reviewers agree that the paper deserves to be published after addressing the raised comments. 

Please submit your revised manuscript by October 25th, 2022. If you need more time than this to complete your revisions, please reply to this message or contact the journal office at plosone@plos.org. Please include the following items when submitting your revised manuscript:A rebuttal letter that responds to each point raised by the academic editor and reviewer(s). You should upload this letter as a separate file labeled 'Response to Reviewers'.A marked-up copy of your manuscript that highlights changes made to the original version. You should upload this as a separate file labeled 'Revised Manuscript with Track Changes'.An unmarked version of your revised paper without tracked changes. You should upload this as a separate file labeled 'Manuscript'.If applicable, we recommend that you deposit your laboratory protocols in protocols.io to enhance the reproducibility of your results. Protocols.io assigns your protocol its own identifier (DOI) so that it can be cited independently in the future. For instructions see: https://journals.plos.org/plosone/s/submission-guidelines#loc-laboratory-protocols. Additionally, PLOS ONE offers an option for publishing peer-reviewed Lab Protocol articles, which describe protocols hosted on protocols.io. Read more information on sharing protocols at https://plos.org/protocols?utm_medium=editorial-email&utm_source=authorletters&utm_campaign=protocols.

We look forward to receiving your revised manuscript.

Kind regards,

Jaume Garcia-Segarra

Academic Editor

PLOS ONE

Journal Requirements:

a) Did participants provide their written or verbal informed consent to participate in this study?

"This work was conducted as part of the Health research programme of the Department of Social Sciences at Wageningen University and Research (WUR). The study itself was funded through a Research Costs Grant for postdoctoral researchers by the Wageningen School of Social Sciences (WASS grant number 21-046)."

"This work was conducted as part of the Health research programme of the Department of Social Sciences at Wageningen University and Research (WUR). The study itself was funded through a Research Costs Grant for postdoctoral researchers by the Wageningen School of Social Sciences (WASS grant number 21-046)."

"This work was conducted as part of the Health research programme of the Department of Social Sciences at Wageningen University and Research (WUR). The study itself was funded through a Research Costs Grant for postdoctoral researchers by the Wageningen School of Social Sciences (WASS grant number 21-046)."

Reviewers' comments:

Reviewer's Responses to Questions

**Comments to the Author**

1. Is the manuscript technically sound, and do the data support the conclusions?

Reviewer #1: Partly

Reviewer #2: Yes

2. Has the statistical analysis been performed appropriately and rigorously? 

Reviewer #1: Yes

Reviewer #2: Yes

3. Have the authors made all data underlying the findings in their manuscript fully available?

Reviewer #1: Yes

Reviewer #2: Yes

4. Is the manuscript presented in an intelligible fashion and written in standard English?

Reviewer #1: Yes

Reviewer #2: Yes

5. Review Comments to the Author

Reviewer #1: The argument is logically presented and it is pleasant to read. In general, I think this paper addressed an important and practical issue.

Introduction

It would be great if the authors may provide more information on cultivation theory. From past studies, were there any conditions that may limit its applicability?

I may have missed the argument. While I can follow why the showing of the photos may influence the descriptive norms, how could injunctive norms be altered when people do not know if their significant others believed that the participants should follow the behaviors displayed in the photos?

Method and result

It is just my speculation. Concerning COVID-19, it seemed that it was less likely that people followed what others did due to the lack of information (because they knew what is the appropriate behavior). If they were driven by social pressure, would the social environment they did the experiment (e.g., whether there were other co-actors) be more relevant to their behaviors (c.f., photo task) -- unless the photos reinforced the injunctive norms? Given that this is an online experiment, would participants care about what other participants (they never met) think (injunctive norms) – they are not their significant others either?

Discussion

I think the discussion is adequate and it paves the way for further investigations. I think it may be quite informative if we also know why did participants stop in the online task given that there was no penalty for violating the physical distance rule.

Reviewer #2: a) Abstract

(i) The overview of the structured abstract was well presented.

(ii) The limitations should be included in the conclusion part of the abstract.

b) Introduction

(i) The importance of the research was stated, the introduction makes all the arguments on the media influence of behavior.

(ii) The study should be guided by clearly itemized research questions/hypotheses. It is not enough to present the questions as items on Table 2. The research questions can be expanded on the questionnaire to achieve each study objective.

c) Method

(i) What is the motivation for selecting the “between-subject design” over other types of studies to address the questions the research hopes to answer?

(ii) In the exclusion criteria for analysis, is it possible for the participants to hide their awareness level by not identifying the aims of the study just to be part of the study? How do the authors account for such a subjective bias?

(iii) The methods section was clearly outlined, and does not require further input.

d) Data Analysis

(i) The appropriate analysis was used to present the results from the data collection process.

(ii) The results were clearly presented and explained.

(iii) The results section does not need further improvement. The results are clearly and lucidly presented.

e) Discussion

(1) The authors can improve the discussion by presenting their impacts using the hypotheses in sequence. Show the implication of the findings on the hypotheses sequentially while using the references to buttress each point.

Chaveesuk, S., Khalid, B., & Chaiyasoonthorn, W. (2022). Continuance intention to use digital payments in mitigating the spread of COVID-19 virus. International Journal of Data and Network Science, 6(2), 527–536. https://doi.org/10.5267/j.ijdns.2021.12.001

Chaveesuk, S., Khalid, B., & Chaiyasoonthorn, W. (2021). Digital payment system innovations: A marketing perspective on intention and actual use in the retail sector. Innovative Marketing, 17(3), 109–123. https://doi.org/10.21511/im.17(3).2021.09

(2) The theoretical and managerial implications should be clearly and logically outlined.

f) Conclusion

(i) The authors should add a recommendations in the conclusion and also study limitations.

g) Title

(i) The title is appropriate and expansive enough to support the ideas that the manuscript represents.

(ii) The title is fine as it is.

(iii) The title needs no further improvement.

6. PLOS authors have the option to publish the peer review history of their article (what does this mean?). If published, this will include your full peer review and any attached files.

Reviewer #1: No

Reviewer #2: No

---

## [Author Response · Author response to Decision Letter 0]

11 Oct 2022

Journal Requirements and Reviewers’ comments: 

Journal Requirements

Point 1: Please ensure that your manuscript meets PLOS ONE's style requirements, including those for file naming. The PLOS ONE style templates can be found at 

Response: We have gone through the above-mentioned guidelines and made several small changes so that our manuscript meets the PLOS ONE style requirements.

Point 2: Please amend your current ethics statement to address the following concerns:

a) Did participants provide their written or verbal informed consent to participate in this study?

Response: Participants provided written consent. We have added this to our methods section as well as to the ethics statement at the bottom of the manuscript.

Point 3: Thank you for stating the following financial disclosure: 

"This work was conducted as part of the Health research programme of the Department of Social Sciences at Wageningen University and Research (WUR). The study itself was funded through a Research Costs Grant for postdoctoral researchers by the Wageningen School of Social Sciences (WASS grant number 21-046)."

Response: As per your request, we have included the amended Role of Funder statement in our cover letter, stating: "The funders had no role in study design, data collection and analysis, decision to publish, or preparation of the manuscript." We thank you in advance for making the required changes on our behalf.

Point 4: Thank you for stating the following in the Acknowledgments Section of your manuscript: 

"This work was conducted as part of the Health research programme of the Department of Social Sciences at Wageningen University and Research (WUR). The study itself was funded through a Research Costs Grant for postdoctoral researchers by the Wageningen School of Social Sciences (WASS grant number 21-046)."

"This work was conducted as part of the Health research programme of the Department of Social Sciences at Wageningen University and Research (WUR). The study itself was funded through a Research Costs Grant for postdoctoral researchers by the Wageningen School of Social Sciences (WASS grant number 21-046)."

Response: As per your request, we have removed all funding information from the acknowledgements section. This section now only reads “We would like to thank the survey participants for their contributions.” The funding statement as it is presented above is correct and may be used in its current form in the online submission form. We thank you in advance for making the required changes on our behalf.

Point 5: Please review your reference list to ensure that it is complete and correct. If you have cited papers that have been retracted, please include the rationale for doing so in the manuscript text, or remove these references and replace them with relevant current references. Any changes to the reference list should be mentioned in the rebuttal letter that accompanies your revised manuscript. If you need to cite a retracted article, indicate the article’s retracted status in the References list and also include a citation and full reference for the retraction notice. 

Response: We have checked all the references, and included two additional citations (one in the introduction section and one in the discussion section), based on the comments of the reviewers (#13 and #33 in the reference list).

Review Comments to the Author

Reviewer #1: The argument is logically presented and it is pleasant to read. In general, I think this paper addressed an important and practical issue.

Response: We thank the reviewer for the compliments and for the time and effort put into reviewing our manuscript. We have considered each suggestion with care. Changes to the revised manuscript are highlighted using the "Track Changes" function in Microsoft Word. 

Point 2: Introduction

It would be great if the authors may provide more information on cultivation theory. From past studies, were there any conditions that may limit its applicability?

Response: We agreed that our introduction would benefit from a more detailed description of the factors influencing the effectivity of media exposure in regards to shaping perceptions and behaviors. Past studies have shown that the factors ‘consistency’ and ‘exposure frequency’ are important in limiting/facilitating its effectivity/applicability. Therefore, we have now added the following in the introduction section on page 4, lines 55-57:

“The consistency of the message, together with exposure frequency, are important factors affecting the exact influence of such media exposure – the reason why the storytelling function of television is so powerful [13].”

Point 3: I may have missed the argument. While I can follow why the showing of the photos may influence the descriptive norms, how could injunctive norms be altered when people do not know if their significant others believed that the participants should follow the behaviors displayed in the photos?

Response: We included the argumentation behind why the showing of the photos may influence injunctive norms on page 5, lines 78-85:

“Second, people may “misperceive” the prevailing injunctive norm – a perception about what should be done according to others – and form (mistaken) beliefs about others’ (dis)approval of certain behavior [6]. Presumably, exposing people to images of others’ non-adherence (such as breaking physical distancing rules) may give people the idea that they have license to break the rules themselves, as this behavior to them seems to be socially acceptable, or at least have limited social sanctions (such as exclusion) [20]. Vice versa, an observation that other people mainly adhere to the communicated rules may lead to the perception that non-adherence to these rules is deviant in nature and will be (socially) sanctioned [14].”

The underlying assumption here is that participants would reason from their observations about the images to their perceptions and behaviors in the ensuing experimental situation (e.g., the behavioral task). In the discussion, page 22, lines 407-413, we have added the following sentences to suggest that participants may not have made this additional deliberative step:

“Reasoning from social norm theory [6], it is known that people typically conform to injunctive norms to build and maintain social relationships with relevant others, thereby increasing (limiting) the chance of social approval (sanctioning) [20]. Inferring injunctive norms thus requires participants to make an inference from the situations as depicted in the images to the situations in the ensuing experimental tasks. Given the non-significant results of the injunctive norm measure, it is a distinct possibility that participants did not engage in this kind of deliberation about potential social approval or disapproval after having been exposed to the images.” 

Point 4: Method and result

It is just my speculation. Concerning COVID-19, it seemed that it was less likely that people followed what others did due to the lack of information (because they knew what is the appropriate behavior). If they were driven by social pressure, would the social environment they did the experiment (e.g., whether there were other co-actors) be more relevant to their behaviors (c.f., photo task) -- unless the photos reinforced the injunctive norms? Given that this is an online experiment, would participants care about what other participants (they never met) think (injunctive norms) – they are not their significant others either?

Response: Thank you for suggesting this line of reasoning. In response, we have included the following explanation in the discussion section on page 22, lines 414-422:

“Another possible explanation from social norm theory pertains to the reference group as described in the descriptive and injunctive social norm measures of this study. The reference group was specified as ‘other participants of this research’, but it is questionable whether people would have sufficiently identified with the people in this reference group, and may therefore not have cared enough about these individuals’ behaviors or their judgments (i.e. what they do or (dis)approve of; related to hypothesis 2a and 2b) to let it influence their own behavior [33]. Instead, the social environment in which participants performed the experiment (e.g., alone or in the presence of others) may have influenced their perceptions and behaviors. A question for further research would then be how the social environment interacts with the exposure manipulation (i.e. the photo task).”

Point 5: Discussion

I think the discussion is adequate and it paves the way for further investigations. I think it may be quite informative if we also know why did participants stop in the online task given that there was no penalty for violating the physical distance rule.

Response: The reviewer is correct to point out that violating the rules in the online behavioral task did not have any negative (monetary) consequences. In the discussion, we acknowledge “[..] that the included hypothetical behavioral measures may not have been adequate proxies for the behavior we were actually interested in, as none of them involved a real threat of infection.” (page 24-25, lines 475-477).

Given the absence of negative monetary incentives, one may expect that subjects never follow the rules in our online behavioral task. However, previous research involving a similar rule-following task, on which our task is based, finds that a majority of subjects follows the rules when this rule is explicitly communicated, as is the case in our design [26, 28]. Both our mean ‘waiting time’ and ‘number of shoppers waited for’ are in line with results from these previous studies. 

Our hypotheses are based on the idea that social norms – irrespective of whether they have been shaped by our experimental treatments – may cause subjects to follow these communicated rules, even when monetary penalties for violating those rules are absent. Indeed, our empirical analysis shows that individual differences in ‘waiting time’ and ‘number of shoppers waited for’ may be explained by perceptions of descriptive and injunctive norms (irrespective of the manipulation), as well as by outcome expectancies (see Tables S2 and S4 in the Supplementary Materials). In our discussion, we chose to discuss this result as a potential explanation for our null result, see page 22, lines 429-434: 

“To illustrate, consider how those who already strongly believed that adherence to physical distancing rules is common and appropriate may have been less easily influenced by photographs depicting other people non-adhering to such rules. This potential explanation of the results is supported by the observation that – irrespective of the manipulation – perceptions of descriptive norms and injunctive norms were correlated with physical distancing behavior (see Supporting information S2 and S4 Tables).” 

Reviewer #2: a) Abstract

(i) The overview of the structured abstract was well presented.

(ii) The limitations should be included in the conclusion part of the abstract.

Response: We appreciate the suggestion, but due to word count limitations, we have opted to focus instead on what we feel is the most plausible explanation of the null results, namely “that a single exposure to such images, especially in the context of COVID-19, is insufficient to affect behavior.”

Point 2: b) Introduction

(i) The importance of the research was stated, the introduction makes all the arguments on the media influence of behavior.

Response: Thank you.

Point 3: (ii) The study should be guided by clearly itemized research questions/hypotheses. It is not enough to present the questions as items on Table 2. The research questions can be expanded on the questionnaire to achieve each study objective.

Response: We now provide an itemized list of hypotheses in the introduction (page 5-6, lines 89-104) and explicitly refer to these hypotheses in the results and discussion section.

Point 4: c) Method

(i) What is the motivation for selecting the “between-subject design” over other types of studies to address the questions the research hopes to answer?

Response: We argue that a between-subject design is more suitable to test our hypotheses than a within-subject design for a variety of reasons. First, within-subject designs are vulnerable to learning or carry-over effects, where exposure to one experimental condition affects subjects’ behaviors in subsequent conditions. Second, there exists an increased risk of experimenter demand effects, as subjects are more likely to guess the research aim when being exposed to multiple experimental conditions. Third, in online experiments with a within-subject design, subjects may experience fatigue or, more likely, boredom when repeatedly answering similar questions, which increases random noise in the data. We included the following in the methods section on page 7, lines 113-114:

“Such a design minimizes the risks of learning, carry-over, experimenter demand, and boredom

 effects.” 

Point 5: (ii) In the exclusion criteria for analysis, is it possible for the participants to hide their awareness level by not identifying the aims of the study just to be part of the study? How do the authors account for such a subjective bias?

Response: We argue that it was not a concern in the present study for the following reasons. First, participants were not aware that how they answered particular questions would impact the extent to which their data was used in the final analysis. Secondly, the payment to participants was not conditional on the inclusion of their data in the final analysis.

Point 6: (iii) The methods section was clearly outlined, and does not require further input.

Response: Thank you.

Point 7: d) Data Analysis

(i) The appropriate analysis was used to present the results from the data collection process.

(ii) The results were clearly presented and explained.

(iii) The results section does not need further improvement. The results are clearly and lucidly presented.

Response: Thank you.

Point 8: e) Discussion

(1) The authors can improve the discussion by presenting their impacts using the hypotheses in sequence. Show the implication of the findings on the hypotheses sequentially while using the references to buttress each point.

Chaveesuk, S., Khalid, B., & Chaiyasoonthorn, W. (2022). Continuance intention to use digital payments in mitigating the spread of COVID-19 virus. International Journal of Data and Network Science, 6(2), 527–536. https://doi.org/10.5267/j.ijdns.2021.12.001

Chaveesuk, S., Khalid, B., & Chaiyasoonthorn, W. (2021). Digital payment system innovations: A marketing perspective on intention and actual use in the retail sector. Innovative Marketing, 17(3), 109–123. https://doi.org/10.21511/im.17(3).2021.09 

Response: To improve the clarity of the manuscript’s discussion we now more clearly mark where particular hypotheses are discussed.

We thank the reviewer for their paper recommendations. We read these with great interest, but unfortunately we fail to see how our findings would relate to the discussions present in the papers. We would welcome the reviewer’s thoughts on how this might be the case.

Point 9: (2) The theoretical and managerial implications should be clearly and logically outlined.

Response: We feel that non-significant findings, like the ones we present in this manuscript, warrant a cautionary stance with regards to making claims about theoretical and managerial implications. We therefore only carefully suggest in the Conclusion section that our results, insofar as they truly are indicative of a weak influence of imagery on adherence behavior, may have implications for how to best communicate about health behaviors. In our opinion, any more speculations about implications are not justified by the present findings.

Point 10: f) Conclusion

(i) The authors should add a recommendations in the conclusion and also study limitations.

Response: As mentioned in response to point 9, we feel that a cautious approach should be taken when drawing conclusions or making recommendations on the basis of non-significant results. Our recommendations are therefore limited to “performing a comparable experiment in which participants are exposed repeatedly to a set of images showing people (non)adhering to a specific behavior in a particular context for a longer period” (page 26, lines 520-522). For a discussion about the methodological limitations of our study design, we refer to the Discussion section (from line 453 onwards).

Point 11: g) Title

(i) The title is appropriate and expansive enough to support the ideas that the manuscript represents.

(ii) The title is fine as it is.

(iii) The title needs no further improvement.

Response: Thank you.

---

## [Decision Letter · Decision Letter 1]

18 Oct 2022

Exposure to images showing (non)adherence to physical distancing rules: Effect on adherence behavior and perceived social norms

PONE-D-22-17082R1

Dear Dr. Raghoebar,

We’re pleased to inform you that your manuscript has been judged scientifically suitable for publication and will be formally accepted for publication once it meets all outstanding technical requirements.

Kind regards,

Jaume Garcia-Segarra

Academic Editor

PLOS ONE

Additional Editor Comments (optional):

Reviewers' comments:

Reviewer's Responses to Questions

**Comments to the Author**

1. If the authors have adequately addressed your comments raised in a previous round of review and you feel that this manuscript is now acceptable for publication, you may indicate that here to bypass the “Comments to the Author” section, enter your conflict of interest statement in the “Confidential to Editor” section, and submit your "Accept" recommendation.

Reviewer #1: All comments have been addressed

Reviewer #2: (No Response)

2. Is the manuscript technically sound, and do the data support the conclusions?

Reviewer #1: Yes

Reviewer #2: No

3. Has the statistical analysis been performed appropriately and rigorously? 

Reviewer #1: Yes

Reviewer #2: No

4. Have the authors made all data underlying the findings in their manuscript fully available?

Reviewer #1: Yes

Reviewer #2: Yes

5. Is the manuscript presented in an intelligible fashion and written in standard English?

Reviewer #1: Yes

Reviewer #2: No

6. Review Comments to the Author

Reviewer #1: (No Response)

Reviewer #2: Apologies, the revised version of the articles has not considered any of the proposed recommendations.

7. PLOS authors have the option to publish the peer review history of their article (what does this mean?). If published, this will include your full peer review and any attached files.

Reviewer #1: No

Reviewer #2: No

---

## [Editor Report · Acceptance letter]

24 Oct 2022

PONE-D-22-17082R1 

Exposure to images showing (non)adherence to physical distancing rules: Effect on adherence behavior and perceived social norms 

Dear Dr. Raghoebar:

I'm pleased to inform you that your manuscript has been deemed suitable for publication in PLOS ONE. Congratulations! Your manuscript is now with our production department. 

Kind regards, 

on behalf of

Dr. Jaume Garcia-Segarra 

Academic Editor

PLOS ONE